# Genome-Wide Identification and an Evolution Analysis of Tonoplast Monosaccharide Transporter (*TMT*) Genes in Seven Gramineae Crops and Their Expression Profiling in Rice

**DOI:** 10.3390/genes14061140

**Published:** 2023-05-24

**Authors:** Mingao Zhou, Xiaoxiao Deng, Yifei Jiang, Guoning Zhou, Jianmin Chen

**Affiliations:** 1Fujian Provincial Key Laboratory of Genetic Engineering for Agriculture, Institute of Biotechnology, Fujian Academy of Agricultural Sciences, Fuzhou 350003, China; 2021202040077@whu.edu.cn (M.Z.); 2021102040039@whu.edu.cn (Y.J.); 2State Key Laboratory of Hybrid Rice, College of Life Sciences, Wuhan University, Wuhan 430072, China; 3The Institute of Subtropical Agriculture, The Chinese Academy of Sciences, Changsha 410125, China; 2017102040003@whu.edu.cn; 4School of Pharmaceutical Sciences, South-Central Minzu University, Wuhan 430074, China; 202121151264@mail.scuec.edu.cn

**Keywords:** cold stress, Gramineae crops, qRT-PCR analysis, rice, tonoplast monosaccharide transporter (*TMT*) gene

## Abstract

The tonoplast monosaccharide transporter (*TMT*) family plays essential roles in sugar transport and plant growth. However, there is limited knowledge about the evolutionary dynamics of this important gene family in important Gramineae crops and putative function of rice *TMT* genes under external stresses. Here, the gene structural characteristics, chromosomal location, evolutionary relationship, and expression patterns of *TMT* genes were analyzed at a genome-wide scale. We identified six, three, six, six, four, six, and four *TMT* genes, respectively, in *Brachypodium distachyon* (Bd), *Hordeum vulgare* (Hv), *Oryza rufipogon* (Or), *Oryza sativa* ssp. *japonica* (Os), *Sorghum bicolor* (Sb), *Setaria italica* (Si), and *Zea mays* (Zm). All TMT proteins were divided into three clades based on the phylogenetic tree, gene structures, and protein motifs. The transcriptome data and qRT-PCR experiments suggested that each clade members had different expression patterns in various tissues and multiple reproductive tissues. In addition, the microarray datasets of rice indicated that different rice subspecies responded differently to the same intensity of salt or heat stress. The Fst value results indicated that the *TMT* gene family in rice was under different selection pressures in the process of rice subspecies differentiation and later selection breeding. Our findings pave the way for further insights into the evolutionary patterns of the *TMT* gene family in the important Gramineae crops and provide important references for characterizing the functions of rice *TMT* genes.

## 1. Introduction

In plants, sugars (including sucrose, monosaccharide, and polyols) participate in plant growth, development, and fruit flavor [1,2]. To date, various types of sugar transporters that transport polyols [3,4], monosaccharides [5], or sucrose [6,7] have been reported, such as the sugars will eventually be exported transporters (SWEET), sucrose transporters (SUT), and monosaccharide transporters (MST) gene families. Among them, the TMT proteins localizing on the tonoplast belong to the *MST* gene family. So far, genome-wide analyses have identified three, three, and six *TMT* genes in *Arabidopsis*, rice, and *Pyrus bretschneideri*, respectively [8,9,10].

The protein structures of the *TMT* gene family members are different from other monosaccharide transporter subfamilies (e.g., STP, PLT, ERD, pGlcT, INT, and XTPH). TMT proteins have a large cytoplasmic ring consisting of about 170 amino terminals between the 6th and 7th transmembrane domains. A large number of phosphorylation sites exist in this large cytoplasmic ring, which are similar to yeast glucose sensors, SNF, and RGT2. Therefore, it has been speculated that TMT protein has a sugar sensing function [11]. Vacuoles from *Arabidopsis attmt1-2*-*3* (*AtTMT1*, -*2*, and -*3* knockout lines) have shown an evident decrease in the absorption capacity of glucose [8]. Expressing the *OsTMT1* gene in vacuoles of the mutant *Arabidopsis* (*attmt1*-*2*) has demonstrated that OsTMT proteins were capable of transporting glucose into vacuoles [9]. In addition, *Arabidopsis* mutant (*attmt1*/*attmt2*) has shown significant decreases in sucrose absorption capacity in vacuoles, which evidenced that AtTMT1 and AtTMT2 may also be involved in sucrose transportation across vacuole membranes [12]. The expression patterns of *TMT* genes in *Arabidopsis* show tissue-specific expression. Among them, *AtTMT1* is mainly expressed in leaves and flowers. *AtTMT2* has been shown to be more dominant in stems and roots, while *AtTMT3* was weakly expressed in all tested tissues [12]. In rice, the expression patterns of *OsTMT1* and *OsTMT2* were very similar and showed an overlapping expression pattern. These two *TMT* genes have high expression levels in rice microtubule sheath cells, parenchyma cells of leaves, and companion cells [8,9]. Most *PbTMTs* are expressed to varying degrees in all tissues; however, *PbTMT5* is highly expressed in stems and flowers [10]. In addition, *PbTMT1* shows relatively uniform expression in various tissues, while *PbTMT2*, *PbTMT3*, and *PbTMT6* are highly expressed in mature leaves. *PbTMT4* is highly expressed in ripe fruit, possibly suggesting a key role in fruit sugar accumulation [10]. The over-expressing *AtTMT1* in *Arabidopsis* has exhibited high growth rate at the seedling stage and increased seed biomass [13]. Wingenter et al. (2010) explained that these physiological changes were due to overexpression of the *TMT* gene, resulting in more efficient sugar sensing amplification and assimilate distribution in plants [13]. In addition, the expression of *AtTMT1* and *AtTMT2* have also been induced by drought, salinity, and chilling stresses [8,11].

The Gramineae class contains a variety of important food crops, with wide distribution areas, which play crucial roles in the global food supply [1,14]. With the development of sequencing technology, whole genome sequencing of rice, wild rice, barley, sorghum, millet, and maize in the Gramineae class have been completed, which provides basic genome data for studying important gene families at the genome-wide level. Based on the public genome sequences of seven Gramineae crops, in this study, we identified the *TMT* genes in *B. distachyon* (Bd) [15], *H. vulgare* (Hv) [16], *S. italica* (Si) [17], *S. bicolor* (Sb) [18], *Z. mays* (Zm) [19], *O. rufipogon* (Or) [20], and *O. sativa* ssp. *japonica* (Os) genomes [21,22]. Subsequently, we analyzed the gene structural features, chromosomal locations, and evolutionary relationships of *TMT* genes at a genome-wide scale, and analyzed the expression profiles of rice *TMT* genes in different tissues at different developmental stages or under different abiotic stresses. The current study provides a theoretical basis for exploring structure characteristics, putative functions, and evolutionary relationships of the *TMT* genes in these important Gramineae crops.

## 2. Materials and Methods

### 2.1. Plant Materials and Treatments

Fifteen rice tissues were collected from “9311” plants grown in a natural environment, during the summer, in Wuhan city (29°58′20″ N, 113°53′29″ E), namely, SC1/2 (seed coat, 3/15 days after flowering), An (anther, 1~3 days before flowering), SO (stigma and ovary, 1~3 days before flowering), ImSe (immature endosperm, 15 days after flowering), Pan5/10/20 (panicles harvested before heading at lengths of 5/10/20 cm), and Car1/3/5/7/10/15/20 (caryopses, 1/3/5/7/10/15/20 days after flowering) [23]. Three biological replicates were produced for every tissue and each biological replicate was collected and pooled together from over 30 plants. To ensure the accuracy of the results, three technical replicates were set for each biological replicate in quantitative real-time PCR (qRT-PCR).

### 2.2. Identification and Phylogenetic Analysis of TMT Genes

In this study, two approaches were adopted to identify TMT proteins in seven tested genomes: hidden Markov model (HMM) and BLAST homology search. Among them, the Bd, Hv, Si, Sb, Zm, and Or genomes were downloaded from Ensembl Plants release 41 (http://plants.ensembl.org/index.html, accessed on 1 December 2022) and the Os (MSU 7.0) genome was obtained from the TIGR database (http://rice.plantbiology.msu.edu, accessed on 1 December 2022) [24]. The HMM profile of the Sugar_tr (PF00083) was obtained from Pfam (http://pfam.xfam.org/, accessed on 1 December 2022) were used to search TMT protein sequences against seven tested crops’ protein sequence datasets with default parameters using HMMER 3.2.1. Then, all identified proteins from the HMM homology searches were researched through the blastP method by local ncbi-blast-2.7.1^+^. Finally, all candidate protein sequences were submitted to the SMART website (http://smart.embl-heidelberg.de/, accessed on 10 December 2022) and the Pfam website (http://pfam.xfam.org/search/sequence, accessed on 10 December 2022) to check the completeness of the Sugar_tr domain.

All identified TMT protein sequences were aligned by ClustalW using PAM protein weight matrix, and a phylogeny tree was produced via the MEGA 6.0 software using the maximum likelihood (ML) method with 1000 bootstrap replicates [1,14,24]. According to clustering results, all *TMT* genes were divided into different clades.

### 2.3. Microsynteny Analysis, Consevered Motifs, and Gene Structure

The collinearity gene pairs of Os with Bd, Hv, Or, Sb, Si, and Zm were analyzed by using the Multiple Collinearity Scan toolkit X version (MCScanX), as described previously [25], and visualized using the ”dual synteny plotter” in TBtools [26]. Twenty conserved motifs were set in the MEME program (http://meme-suite.org/tools/meme, accessed on 20 December 2022) with the following parameters: motif width between 6 and 100, and other default parameters [1]. Genomic structures of *TMT* genes were gained from genomic annotation files. Finally, we used TBtools to visualize the result of the phylogenetic tree, gene structures, and conserved motifs of *TMT* genes [26].

### 2.4. Quantitative Analysis of TMT Genes in Rice

All RNAs were extracted using TRIzol Ragent (Invitrogen, Beijing, China) and reversed to cDNAs using HiScript III 1st Strand cDNA Synthesis SuperMix for qPCR (Cat No. 11141ES60, Yeasen, Shanghai, China). The qRT-PCR reactions (10 μL) were formulated using Hieff UNICON Universal Blue qPCR SYBR Green Master Mix (Cat No. 11184ES08; Yeasen, Shanghai, China), following the manufacturer’s protocol. The primers were designed by Primer 5.0 (Appendix A). All qRT-PCR reactions were conducted on a CFX96 Touch™ Real-Time PCR Detection System (Bio-Rad, Hercules, CA, USA). The actin gene was used as an internal control [14,23] and relative expression values were calculated by the 2^−ΔΔCT^ method based on 3 biological replicates ×3 technical replicates [1,14]. All heatmaps were created using R package (pheatmap).

### 2.5. Expression Analysis of Rice TMT Genes under Cold and Salt Stress

Normalized intensities data of three-leaf-stage shoots and roots of TNG67 (*indica*) and TCN1 (*japonica*) under cold stress (4 °C, GSE57895, 96 microarray datasets) and salt stress (250 mM NaCl treatment, GSE76613, 96 microarray datasets) were download from the Gene Expression Omnibus (GEO, https://www.ncbi.nlm.nih.gov/geo/, accessed on 20 December 2022). The expression changes in *TMT* genes after stress treatment were calculated using the formula: fold change (FC) = average expression amount of the treatment groups/average expression amount of the control groups.

### 2.6. Population Genetic Differentiation Coefficient (Fst, Fixation Index) of Rice TMT Genes

Based on the published rice 3K resequencing data (3K RG 1M GWAS SNP Dataset, all chromosomes, http://iric.irri.org/, accessed on 20 December 2022) including 1770 *indica* rice and 850 *japonica* rice, the Fst values of SNPs in the open reading frame (ORF) of the *TMT* genes identified in this study were calculated using the vcftools software. The average Fst value of all SNPs in the ORF of each gene was recorded as the Fst value of this gene.

## 3. Results

### 3.1. Identification of TMT Genes in Gramineae Crop Genomes

A total of 35 *TMT* genes were identified in seven Gramineae crop genomes, namely, six genes in Bd, three genes in Hv, six genes in Or, six genes in Os, four genes in Sb, six genes in Si, and four genes in Zm (Table 1). The identified *TMT* genes were named *BdTMT*, *HvTMT*, *OrTMT*, *OsTMT*, *SbTMT*, *SiTMT*, and *ZmTMT* followed by a number according to the chromosomal order in each genome, in accordance with the previous TMT study [27,28]. To understand the evolutionary relationships of these *TMT* genes among Gramineae crops, an ML phylogenetic tree was generated using MEGA 6.0. As shown in Table 2 and Figure 1, all TMT proteins were divided into three clades: I, II, and III. Clade III belonged to single-gene clade and had fewer *TMT* genes than those in clades I and II (Figure 1). Further analysis found that no clade III *TMT* gene existed in Hv. A selective pressure analysis showed that all three clades were under negative selective forces. We speculated that the number difference between different species may be caused by genetic duplication events. However, the MCScanX results showed that no gene duplication events were found in all tested species. These results indicated that the ancestors of Gramineous plants may have six *TMT* ancestor genes and that different gramineous species have unequal loss of *TMT* genes during species differentiation, which resulted in the current difference in the number of *TMT* genes.

### 3.2. Collinearity Gene Pairs, Intron/Exon Structure, and Conserved Motifs

The chromosome location results of *TMT* genes showed that the *TMT* genes were unevenly distributed on all chromosomes of the tested species. *TMT* genes were found only in certain segments of some chromosomes (Appendix A). All these segments had good linear relationships between the Os genome and other tested crop genomes (Figure 2). Here, we identified four, two, eight, four, five, and two collinearity gene pairs between Os and Bd, Hv, Or, Sb, Si, and Zm, respectively (Figure 2). There were more collinearity gene pairs between Os and Or, Si, Bd, and Sb than between OS and Hv and Zm, which basically supported the evolutionary distance of the tested species.

In addition, intron/exon structure and conserved motifs of *TMT* genes in Gramineae crops revealed that *TMT* genes were very conserved in different species because they had similar structural features (Figure 3). Genes from the same clade shared similar gene structures and conserved motif organizations, while genes from different clades showed different gene structures and conserved motif organizations. For example, clades I (5–8 exons) and III (5–6 exons) possessed more exons than clade II (1–2 exons). In summary, *TMT* genes were relatively conserved among Gramineae crops, while structural differentiaition had occurred in different clades during the process of evolution, which provided a structural basis for the functional differentiation of *TMT* genes in species. Of course, this speculation needs to be further supported by evidence from gene expression data.

### 3.3. Expression Patterns of TMT Genes in Various Tissues of Rice

Our expression profiling results revealed that rice *TMT* genes showed different expression levels in various tissues at different development stages or in 15 reproductive tissues (Figure 4). OsTMT1 and OsTMT4 showed higher expression levels in most tissues as compared with OsTMT3, OsTMT2, and OsTMT6 (Figure 4A). OsTMT5 showed specific expression of panicle, spike, and pollen (Figure 4A), suggesting that this gene may play roles in these specific tissues.

*OsTMT1* showed the highest expression level in Car5. *OsTMT2* was highly expressed in SC2. *OsTMT3* had the highest expression level in Car15. However, *OsTMT4* displayed the highest expression levels in pan10 and pan20. *OsTMT6* showed relatively high expression levels in pan10 and SC2 (Figure 4B). These clearly differentiated expression profiles further prove that rice *TMT* genes had undergone functional differentiations.

### 3.4. Expression Patterns of TMT Genes in Rice under Cold or Salt Stress

It is well known that salt stress and cold stress are the two main abiotic stresses that affect the normal growth and yield of rice [29,30]. *Indica* and *japonica*, as two subspecies, show different levels of stress tolerance under the same strength of stress pressure [30]. In addition, the roles of rice *TMT* genes in salt stress and cold stress are still unclear.

In this study, the expression changes in rice *TMT* genes under cold and salt stresses were studies (Figure 5). We found that some *TMT* genes in shoots and roots had different transcriptional responses to salt and cold stress, such as *OsTMT4* and *OsTMT1* in shoots and roots under salt stress (Figure 5A), and ^OsTMT2^ in shoots and roots under cold stress (Figure 5B). Under the same stress treatment, most of the *TMT* genes showed similar expression profiles in *indica* and *japonica* rice, but some genes were significantly different in *indica* and *japonica* rice, namely, *OsTMT3* and *OsTMT5* under salt stress (Figure 5A) and *OsTMT3* under cold stress (Figure 5B). The expression of *OsTMT6* was induced in both shoots and roots at 3 h and 24 h after salt stress treatment in two subspecies. Additionally, *OsTMT3* showed different expression changes in *indica* and *japonica* rice after cold stress treatment. This gene was upregulated in the root of TNG67 at 3 h, 24 h, and Re: 24 h, while it was upregulated in the root of TCN1 at only 24 h (Figure 5B).

### 3.5. Fst Values of Rice TMT Genes

Previous studies have reported that an Fst value of 0–0.05 indicates that there is no differentiation among the populations; an Fst value of 0.05–0.15 indicates that the populations are moderately differentiated; an Fst value of 0.15–0.25 indicates that the populations are highly differentiated; an Fst value greater than 0.25 indicates complete differentiation [31,32,33]. In this study, *OsTMT2* was moderately differentiated between *indica* and *japonica* rice populations (0.0772). *OsTMT6* was highly differentiated between *indica* and *japonica* rice populations (0.2476). *OsTMT1*, *OsTMT3*, *OsTMT4*, and *OsTMT5* were completely differentiated between *indica* and *japonica* rice populations (>0.25) (Table 3). In particular, the Fst value of *OsTMT4* was 0.9304, indicating that the gene was extremely differentiated between *indica* and *japonica* rice populations. These results indicated that the rice *TMT* gene family was under different selection pressures in the process of rice subspecies differentiation and later selection breeding.

## 4. Discussion

### 4.1. TMT Gene Lose in Gramineae Crop Genomes during the Process of Evolution

With the development of genome sequencing technology, more and more gene families have been identified in a large number of plants, such as *GH3*, *GT8* and the *AGC* gene family [14,34,35,36]. Benefiting from genome-wide bioinformatics coupled with quantitative analysis, we identified and characterized the *TMT* gene family in Gramineae crops. Several studies have reported that gene duplication events play crucial roles in gene family expansion [5,14,37,38]. Our earlier study also found that duplication events led to the expansion of the rice STP, ERD, and PLT gene families (these all belong to the MST family) [28]. However, no gene duplication event of the *TMT* gene family was found in these seven species. On the contrary, gene lose may have occurred in some Gramineae crops. Clade III was lost in Hv. In addition, we found that there were no direct positive correlations between the number of *TMT* genes and plant genome sizes or whole genome duplication events. For example, the genome size of Hv is 4.79 Gbp with only three *TMT* genes, but the genome size of Os is 382.78 Mbp with six *TMT* genes. Swigoňová reported that *Z. mays* underwent one specific WGD more than other Gramineae plants [39]. However, in this study, *Z. mays* had fewer *TMT* genes than Bd, Or, Os, and Si. Taken together, the *TMT* gene tends to be genetically lost rather than expanded during the differentiation of gramineous species.

### 4.2. Functional Differentiation of Rice TMT Genes Involves Multiple Tissue Development Processes and Cold Stress Responses

Previous studies have shown that genes with the same function have similar gene structures and expression profiles [40,41,42]. To characterize the putative function of *TMT* genes, the gene structures and tissue expression profiles of *TMT* genes in rice were analyzed. As a result, the differential expression profiles of *TMT* genes indicated that the rice *TMT* genes had undergone functional differentiations. Expression profiling results revealed that rice *TMT* genes showed different expression levels in various tissues at different development stages or in 15 reproductive tissues (Figure 4). OsTMT1 and OsTMT4 showed higher expression levels in most tissues than OsTMT3, OsTMT2, and OsTMT6 (Figure 4A). In addition, some genes were highly expressed at a certain stage of rice organ development (e.g., *OsTMT1* in Car7, *OsTMT2* in SC2, and *OsTMT3* in Car15), which demonstrated that these *TMT* genes are involved in rice reproductive development.

Salt and cold are the major abiotic stresses that frequently affect the growth, development, and food yield of crops in many countries [14,43,44]. The mechanism by which *TMT* genes promote sugar transportation or accumulation has been studied extensively [8,42,45], but it is not clear whether *TMT* genes play roles under abiotic stress. In this study, *OsTMT3* showed obvious upregulation under salt stresses. In addition, we found that some *TMT* genes in shoots and roots had different transcriptional responses to salt and cold stress, such as *OsTMT4* and *OsTMT1* in shoots and roots under salt stress (Figure 5A) and *OsTMT2* in shoots and roots under cold stress (Figure 5B). These results revealed that *TMT* genes are associated with salt stress and that the *OsTMT3* gene could be a good candidate gene for rice resistance breeding under cold stress.

### 4.3. Different Selection Pressures May Promote the Expression Differentiations and Functional Differentiations of TMT Genes

In this study, the phylogenetic tree, gene structure, and protein motifs all supported the division of *TMT* genes into three clades. In addition, the expression patterns of the rice *TMT* genes from three clades were varied in various tissues or under abiotic stresses. The Tajima’s D values of the three clades were different, suggesting that the *TMT* genes of the three clades were under different intensities of selection pressure. It is worth noting that the expression profiles of some *TMT* genes in *indica* and *japonica* subspecies were also different. The Fst values of all *TMT* genes were greater than 0.05, and the Fst value of four *TMT* genes were greater than 0.25. These results suggested that *TMT* genes had been severely differentiated in the two *indica* and *japonica* populations, which may be a reason for the differential expressions of rice *TMT* genes in these two subspecies.

## 5. Conclusions

In the present study, a total of 35 *TMT* genes were identified in seven Gramineae crop genomes, and all TMT proteins could be subdivided into three clades. Subsequently, we carried out a systematic bioinformatics analysis at the genome-wide scale, including a phylogenetic analysis, exon/intron structures, conserved motifs, and collinearity relations. Our findings suggested that some *TMT* genes may have been lost in some Gramineae crops. Expression profiling of rice *TMT* genes revealed that rice *TMT* genes may act in multiple tissues and also respond to salt and cold stress. The comprehensive analysis of the *TMT* genes in this study will be useful for further research on biological functions and the evolution of *TMT* genes in Gramineae crops.

## Figures and Tables

**Figure 1 genes-14-01140-f001:**
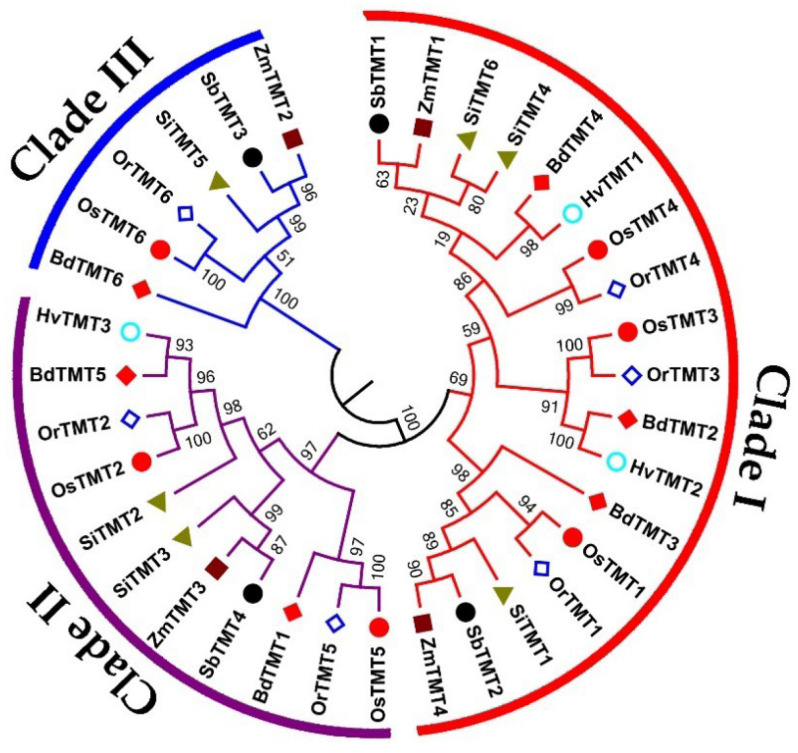
A maximum likelihood (ML) phylogeny tree of TMT protein sequences from *O. sativa* ssp. *japonica* (Os), *O. rufipogon* (Or), *H. vulgare* (Hv), *S. bicolor* (Sb), *B. distachyon* (Bd), *S. italica* (Si), and *Z. mays* (Zm). The genes in the red, purple, and blue range belong to clades Ⅰ, Ⅱ, and Ⅲ, respectively.

**Figure 2 genes-14-01140-f002:**
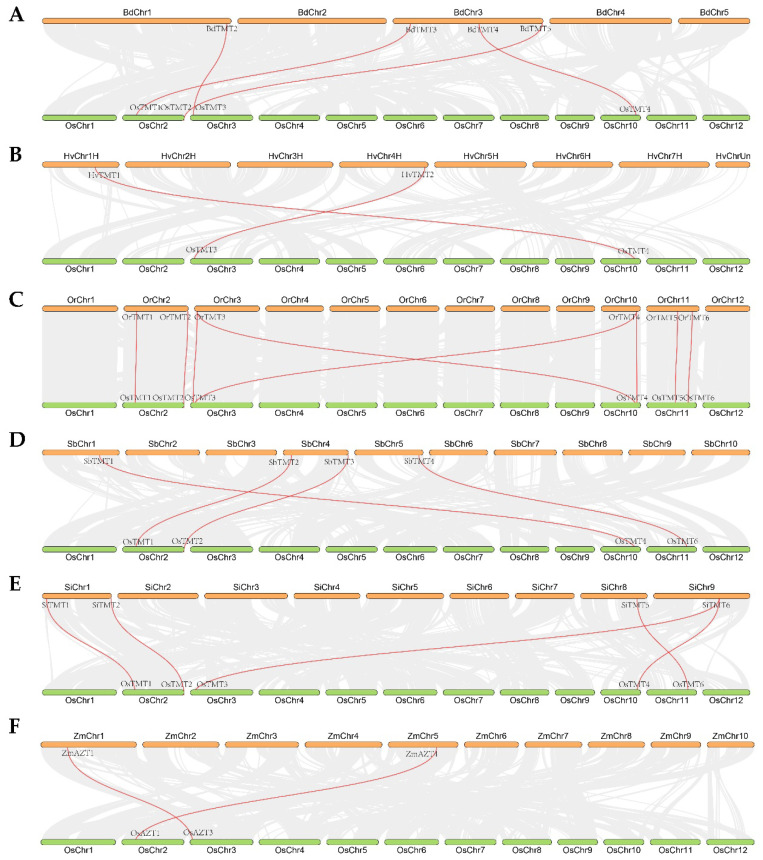
Collinearity relationships of *TMT* genes between *O. sativa* ssp. *japonica* (Os) and *B. distachyon* (Bd), *H. vulgare* (Hv), *O. rufipogon* (Or), *S. bicolor* (Sb), *S. italica* (Si), and *Z. mays* (Zm), respectively: (**A**) Os vs. Bd; (**B**) Os vs. Hv; (**C**) Os vs. Or; (**D**) Os vs. Sb; (**E**) Os vs. Si; (**F**) Os vs. Zm.

**Figure 3 genes-14-01140-f003:**
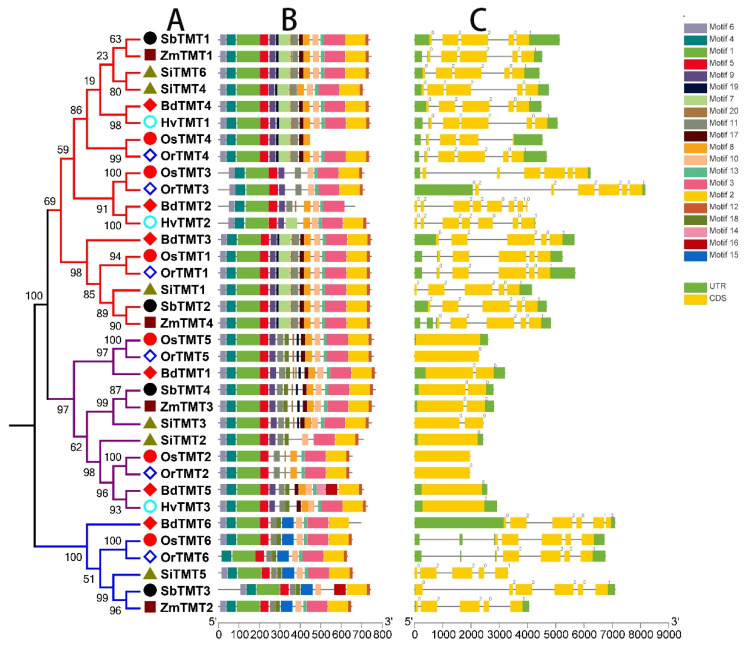
Phylogenetic tree (**A**), motif compositions (**B**), and exon/intron structure (**C**) of the *TMT* genes in seven Gramineae crops. Different branches in the phylogenetic tree represent different clades. The relative lengths of proteins and genes can be estimated by using the gray bars. Untranslated regions (UTRs), exons, and introns are represented by green blue boxes, yellow boxes and gray lines, respectively.

**Figure 4 genes-14-01140-f004:**
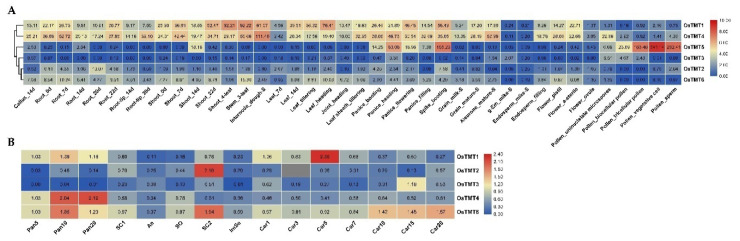
(**A**,**B**) Expression profiles of rice *TMT* genes in different tissues; (**B**) SC1/2 (seed coat, 3/15 days after flowering), An (anther, 1~3 days before flowering), SO (stigma and ovary, 1~3 days before flowering), ImSe (immature endosperm, 15 days after flowering), Pan5/10/20 (panicles harvested before heading at lengths of 5/10/20 cm), and Car1/3/5/7/10/15/20 (caryopses, 1/3/5/7/10/15/20 days after flowering).

**Figure 5 genes-14-01140-f005:**
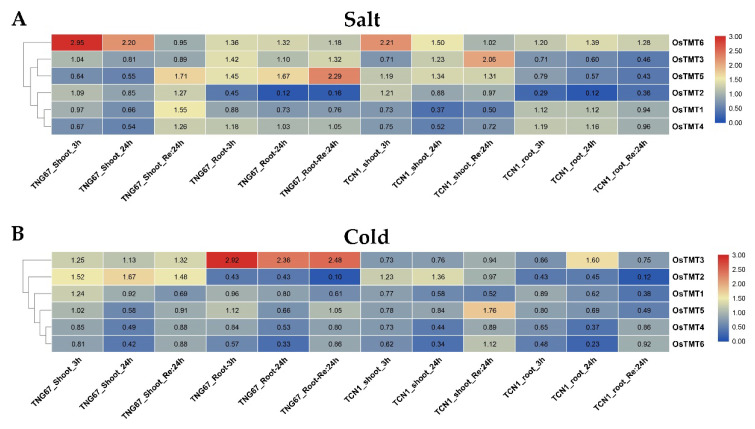
Expression changes in the rice 3-leaf seedling under salt (**A**) and cold (**B**) stress. 3 h, 24 h, and Re: 24 h mean 3 h after treatment, 24 h after treatment, and recovering 24 h after termination of treatment.

**Table 1 genes-14-01140-t001:** The detailed information of tonoplast monosaccharide transporter (*TMT*) genes in seven Gramineae crops.

Name	Chr	Chr.start	Chr.end	Protein ID	Gene ID	Clade	Total
BdTMT3	BdChr3	6859522	6865177	KQJ94129	BRADI_3g08690v3	I	18
BdTMT4	BdChr3	34222123	34226613	KQJ97606	BRADI_3g32210v3	I	
HvTMT1	HvChr1H	385752912	385757976	HORVU1Hr1G052040.1	HORVU1Hr1G052040	I	
OrTMT1	OrChr2	6762121	6767808	ORUFI02G09610.1	ORUFI02G09610	I	
OrTMT4	OrChr10	18702549	18707230	ORUFI10G18200.1	ORUFI10G18200	I	
OsTMT1	OsChr2	7273033	7278272	LOC_Os02g13560.1	LOC_Os02g13560	I	
OsTMT4	OsChr10	21048271	21052794	LOC_Os10g39440.2	LOC_Os10g39440	I	
SbTMT1	SbChr1	59989927	59995061	EER94578	SORBI_3001G312900	I	
SbTMT2	SbChr4	8829374	8834055	KXG29849	SORBI_3004G099300	I	
SiTMT1	SiChr1	2517685	2521836	KQL28009	SETIT_016433mg	I	
SiTMT4	SiChr8	330126	334877	KQK93278	SETIT_026033mg	I	
SiTMT6	SiChr9	40109050	40113465	KQK90140	SETIT_034411mg	I	
ZmTMT4	ZmChr5	155133354	155138175	Zm00001d016274_P006	Zm00001d016274	I	
ZmTMT1	ZmChr1	86130014	86134528	Zm00001d029762_P004	Zm00001d029762	I	
BdTMT2	BdChr1	73328389	73332373	PNT78271	BRADI_1g76540v3	I	
HvTMT2	HvChr4H	623189185	623193457	HORVU4Hr1G082770.1	HORVU4Hr1G082770	I	
OrTMT3	OrChr3	1375312	1383473	ORUFI03G02040.1	ORUFI03G02040	I	
OsTMT3	OsChr3	1631964	1638192	LOC_Os03g03680.1	LOC_Os03g03680	I	
BdTMT1	BdChr1	25183013	25186217	KQK16630	BRADI_1g29600v3	II	11
BdTMT5	BdChr3	59349777	59352344	KQK02174	BRADI_3g60756v3	II	
HvTMT3	HvChr6H	582642918	582645833	HORVU6Hr1G095020.1	HORVU6Hr1G095020	II	
OrTMT2	OrChr2	33807325	33809283	ORUFI02G40100.1	ORUFI02G40100	II	
OrTMT5	OrChr11	16365550	16367823	ORUFI11G14180.1	ORUFI11G14180	II	
OsTMT2	OsChr2	35779430	35781388	LOC_Os02g58530.1	LOC_Os02g58530	II	
OsTMT5	OsChr11	16510126	16512724	LOC_Os11g28610.1	LOC_Os11g28610	II	
SbTMT4	SbChr10	60923801	60926597	EER90435	SORBI_3010G276100	II	
SiTMT3	SiChr4	4085784	4088214	KQL09535	SETIT_008056mg	II	
SiTMT2	SiChr1	42010600	42013022	KQL32269	SETIT_016489mg	II	
ZmTMT3	ZmChr5	66605228	66608036	Zm00001d014872_P001	Zm00001d014872	II	
BdTMT6	BdChr4	13625513	13632606	KQJ87747	BRADI_4g13310v3	III	6
OrTMT6	OrChr11	24071394	24078151	ORUFI11G21610.1	ORUFI11G21610	III	
OsTMT6	OsChr11	24184610	24191330	LOC_Os11g40540.2	LOC_Os11g40540	III	
SbTMT3	SbChr5	67550181	67557281	EES08863	SORBI_3005G191900	III	
SiTMT5	SiChr8	34519780	34523079	KQK95449	SETIT_027425mg	III	
ZmTMT2	ZmChr4	6013141	6017194	Zm00001d048823_P007	Zm00001d048823	III	

Note: *B. distachyon* (Bd), *H. vulgare* (Hv), *O. rufipogon* (Or), *O. sativa* ssp. *japonica* (Os), *S. bicolor* (Sb), *S. italica* (Si), and *Z. mays* (Zm). Chr means chromosome.

**Table 2 genes-14-01140-t002:** Gene numbers of *TMT* genes in seven tested species and Tajima’s values of the three clades.

Species	I	II	III	Total
Hv	2	1	0	3
Bd	3	2	1	6
Os	3	2	1	6
Or	3	2	1	6
Si	3	2	1	6
Sb	2	1	1	4
Zm	2	1	1	4
Total	18	11	6	35
Tajima’s D	−0.60342	−0.60423	−0.18601	

Note: *H. vulgare* (Hv), *B. distachyon* (Bd), *O. sativa* ssp. *japonica* (Os), *O. rufipogon* (Or), *S. italica* (Si), *S. bicolor* (Sb), and *Z. mays* (Zm).

**Table 3 genes-14-01140-t003:** Fst values of rice *TMT* genes.

Genes	Chr	Chr.start	Chr.end	SNP Number	Average Fst
OsTMT1	2	7273033	7278272	12	0.2952
OsTMT2	2	35779430	35781388	3	0.0772
OsTMT3	3	1631964	1638192	6	0.5067
OsTMT4	10	21048271	21052794	2	0.9304
OsTMT5	11	16510126	16512724	3	0.4581
OsTMT6	11	24184610	24191330	20	0.2476

## Data Availability

All data generated or analyzed during this study are included in this published article.

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
