# Peer review of "Genome-Wide Identification and an Evolution Analysis of Tonoplast Monosaccharide Transporter (TMT) Genes in Seven Gramineae Crops and Their Expression Profiling in Rice"

_genes, 2023, doi:10.3390/genes14061140_

Round 1
Reviewer 1 Report
The article “Genome-wide Identification of Tonoplast Monosaccharide Transporter (TMT) Genes in Seven Gramineae Crops and Their Expression Profiling in Rice” is well written described. Tonoplast Monosaccharide Transporter (TMT) family plays essential roles in sugar transport and plant growth. The comprehensive analysis of the TMT genes will be useful for further research on biological functions and the evolution of TMT genes in Gramineae crops. I have just a few comments/suggestions.
1. Mention the novelty statement of your article.
2. What are the criteria for the selection of seven Gramineae Crops?
3. Add the recent studies related to this study.
4. Match the references and their citation inside the text and improve according to the journal format.
5. Format the manuscript headings according to journal instructions and check the English grammar.
Author Response
Dear reviewer,
Thank you for your suggestion. Please see the attachment
Best wishes,
Mingao Zhou

Reviewer 2 Report
1. Authors should revise the Abstract section of the manuscript and write 2-3 line of result.
2. In the Keyword section, qRT-PCR analysis instead only writes qRT-PCR
3. Add recent references in the Introduction section of the Ms.
4. In the Introduction section is too short add 2 more paragraph.
5. In the material and methods section, provide the details information of sample collection (date, time etc.)
6. Why authors are using Mega 6.0 why not latest software.
7. In the phylogenetic tree you have mention different symbol kindly describe in the legend section.
8. The Discussions section need more references, integrated data and score subject wise.
9. There are grammatical mistake throughout the manuscript kindly check and correct it.
10. Add few recent article.
Minor editing of English language required
Author Response

(The authors gave the same response as above.)

Reviewer 3 Report
This is a well-written manuscript. In this study, the authors identified a total of 35 TMT genes in seven Gramineae crops at the genome-wide scale using a systematic bioinformatics analysis, including phylogenetic analysis, exon/intron structures, conserved motifs, collinearity relations. The results indicated that some TMT genes may have been lost in some Gramineae crops. The authors also carried out expression profiling of rice TMT genes, which revealed that rice TMT genes may act in multiple tissues and also respond to salt and cold stress. The data is new and abundant. The results are helpful for further research on biological functions and the evolution of TMT genes in Gramineae crops. However, there are still some minor issues needed to be modified before acceptation.
1) Figure 2, Page 7: The authors should mention A, B, C, D, E, and F in the Figure Captions.
2) Figure 3, Page 8: The authors need to label A, B, and C, respectively, in the figure.
3) Figure 4, Page 9: The authors should mention A and B in the Figure Captions.
4) Figure 5, Page 9: In the Figure Captions, “…6h after treatment…” should “…24h after treatment …”, am I right?
5) Figure S1: The authors need to redraw Figure S1-G.
Author Response

(The authors gave the same response as above.)
